# Pupal Cues Increase Sperm Production but Not Testis Size in an Insect

**DOI:** 10.3390/insects12080679

**Published:** 2021-07-28

**Authors:** Junyan Liu, Xiong Z. He, Xia-Lin Zheng, Yujing Zhang, Qiao Wang

**Affiliations:** 1School of Agriculture and Environment, Massey University, Palmerston North 4410, New Zealand; J.Liu4@massey.ac.nz (J.L.); X.Z.He@massey.ac.nz (X.Z.H.); 2Guangxi Key Laboratory of Agric-Environment and Agric-Products Safety, National Demonstration Centre for Experimental Plant Science Education, College of Agriculture, Guangxi University, Nanning 530004, China; zhengxialin329@163.com (X.-L.Z.); helena9077@163.com (Y.Z.)

**Keywords:** spermatogenesis, sperm competition, testes, socio-sexual environment

## Abstract

**Simple Summary:**

Animals adjust their resource allocation strategies to maximize their reproductive benefit under dynamic socio-sexual environments. For example, male insect adults increase their testicular investment with the perceived increase of rivals to gain a competitive advantage in fathering offspring. To date, it is not clear whether insect pupae, which do not feed and crawl, can fine-tune their investment in sperm and testis size according to their social-sexual settings. This knowledge is vital to understanding how male insects respond to their surroundings experienced at different life stages. Using a moth which produces both fertile and unfertile sperm, we demonstrated for the first time that after detecting cues from conspecific pupae regardless of sex, male pupae increased production of both types of sperm at the same rate but kept testis size unchanged. Because most morphological traits are formed during the larval stage in insects, testis size may be fixed after pupation, allowing little room for the pupae to adjust testis size with social changes. Like adults, male pupae with fully grown testes have sufficient resources to produce more sperm of both types according to the perceived increase of sperm competition risk.

**Abstract:**

Theoretic and empirical studies show that social surroundings experienced by male insects during their larval or adult stage can influence their testicular investment in diverse ways. Although insect pupae do not feed and crawl, they can communicate using sex-specific and/or non-sex specific cues. Yet, it is unknown, in any insect, whether and how male pupae can fine-tune their resource allocation to sperm production and testis size in response to socio-sexual environments. We investigated this question using a moth, *Ephestia kuehniella*, which produces fertile eupyrene sperm and unfertile apyrene sperm. We held male pupae individually or in groups with different sex ratios, and dissected adults upon eclosion, measured their testis size, and counted both types of sperm. We demonstrated that after exposure to conspecific pupal cues regardless of sex, male pupae increased production of eupyrenes and apyrenes at the same rate but kept testis size unchanged. We suggest that testis size is fixed after pupation because most morphological traits are formed during the larval stage, allowing little room for pupae to adjust testis size. Like adults, male pupae with fully grown testes have sufficient resources to produce more sperm of both types according to the perceived increase in sperm competition risk.

## 1. Introduction

Animals adjust their resource allocation strategies to maximize their reproductive fitness in response to socio-sexual environments [1,2,3]. For example, male animals may invest more in sperm after they detect the presence of rivals to gain an advantage in sperm competition [2,4,5,6,7,8,9,10]. In insects, males fine-tune their sperm investment in response to sex specific cues experienced during the adult stage [10,11,12,13,14,15,16] or non-sex specific cues during the larval stage [17,18,19,20,21]. Although insect pupae do not feed and crawl, they can communicate with each other using species-specific acoustic [22,23,24,25,26,27,28,29] or chemical cues [30,31,32,33,34]. Furthermore, female pupae can release sex pheromones [35,36,37,38]. These findings suggest that male pupae should be able to detect conspecific pupal cues representing the density and sex ratio of the local population, and thus future sperm competition risk. Yet, prior to the current study, nothing is known about whether and how insect pupae can adjust their sperm production in response to these cues.

Testes are a sperm production organ and their relative size or mass may be an indicator of sperm production. Evidence shows that male insect larvae, in the growth and development stage, can adjust their testis size in response to conspecific larval cues, regardless of sex. For example, with the increase in larval density, testis size increases in some species, suggesting an increase of sperm production (sperm were not counted though) [17,39,40]. In a study where testis size is measured and sperm are counted [19], the male larvae exposed to larval cues, regardless of sex, produce smaller testes but more fertile sperm. These discoveries suggest that in response to their social environment, male larvae are able to dedicate varying portions of testis volumes to spermatogenesis and other functions [41], resulting in potential trade-offs between traits of different functions [19,42,43,44]. However, there is no report that insects can alter their testis size in response to the socio-sexual environment experienced at the adult stage. This may be because most resource allocation to traits making up the adult body takes place during growth and development [42,43,45,46,47,48], leaving little room for adults to change their testis morphology. To date, it is not clear whether insect pupae can alter their testis size after exposure to different socio-sexual environments. 

Here, we used a polygamous moth, *Ephestia kuehniella*, as a model to investigate whether and how the socio-sexual environment during the pupal stage affects male investment in testis size and sperm production. Adults of this species do not feed so they acquire all resources via larval feeding [49,50]. The pupal stage lasts about eight days [19,51,52], during which time, females emit sex pheromones [36]. Like most lepidopterans [53], *E. kuehniella* males produce two types of sperm, larger nucleated eupyrenes during the larval and pupal stages, and smaller anucleated apyrenes during the pupal stage [54]. After mating, both types of sperm migrate to the sperm storage site (spermatheca) but only eupyrenes can fertilize eggs. Apyrenes may function to delay female remating [55,56], protect eupyrenes in the female reproductive tract [57], or enable eupyrenes to migrate to the spermatheca [58]. More recent studies suggest that the role of apyrenes may be completed after both types of sperm arrive at the spermatheca [59,60]. The apyrene-to-eupyrene ratio remains consistent under food shortage during the larval stage [61] or environmental stress during the larval [62] and pupal stages [63]. However, *E. kuehniella* males increase the ratio after detecting rival cues during the adult stage [10] or reduce it following exposure to larval cues during the larval stage [19]. So far, it is still unclear whether the socio-sexual environment during the pupal stage affects the sperm production ratio.

Based on the theoretic framework and empirical evidence outlined above, we hypothesize that male pupae kept together with other male pupae should grow larger testes and produce more sperm with higher apyrene-eupyrene ratio than those maintained individually or with female pupae. To test this prediction, we individually reared hundreds of larvae under the same condition, starting from neonate larvae. We then transferred newly pupated pupae to experimental arenas and held male pupae individually or in groups with different sex ratios. Upon adult eclosion, we dissected them, measured their testis size, and counted both types of sperm. This is the first study to examine whether and how male insects adjust their testicular investment in response to their socio-sexual environment experienced during the pupal stage. 

## 2. Materials and Methods

### 2.1. Insects

We established a laboratory colony of *E. kuehniella* from thousands of larvae collected at Turks’ Poultry, Foxton, New Zealand. We raised these larvae with their original food until adult eclosion in the laboratory. To standardize the colony, we randomly selected and confined about 300 newly eclosed adults (approx. 1:1 sex ratio; females with an ovipositor and males with a pair of claspers at the end of abdomen) in a transparent plastic cage (28 cm in length and width and 24 cm in height), lined with porous plastic sheets on the bottom for oviposition. We then randomly allocated 200 resultant neonate larvae to each of the 10 transparent plastic cylinders (8 cm in diameter and 10 cm height), each filled with 50 g artificial diet (ad libitum) comprising of a 3.0:10.0:43.5:43.5 mixture of yeast, glycerine, maize meal, and whole meal wheat flour, respectively. We covered the cylinder with a lid. We made a hole (3 cm diameter) in the middle of the lid and covered it with two layers of cloth mesh (2.8 apparatus per mm^2^) for ventilation. 

To generate an experimental line, we randomly collected 1000 neonate larvae produced by adults from the above cylinders and reared them individually in 2.0-mL micro-centrifuge tubes, each with 0.25 g artificial diet for food and a ventilation hole in the lid made by an insect pin. We observed their pupation daily after the larvae reached the final (sixth) instar. The breeding colony and experimental line were kept and all experiments conducted at 25 ± 1 °C and 60 ± 10% RH with a photoperiod of 10:14 h (dark:light). 

### 2.2. Experimental Setup and Data Collection

We randomly selected newly pupated pupae (male pupae with visible reddish testes in the abdomen) from the experimental line and transferred them into glass vials (2 cm in diameter and 7.5 cm height) to create three treatments (Figure 1): (1) one male pupa in a vial (1M), (2) six male pupae in a vial (6M), and (3) one male pupa and five female pupae in a vial (1M5F). Pupae in treatments (2) and (3) were in close contact with each other. We plugged the glass vial opening with cotton wool and monitored adult emergence daily six days after transfer. All pupae from the vials successfully emerged. Immediately after eclosion, we individually transferred newly emerged male adults into micro-centrifuge tubes, clearly labelled them and placed them at −20 °C in a freezer. We considered all emerged males as replicates, i.e., the male from each 1M vial, the male from each 1M5F vial, and all six males from each 6M vial. In total, we obtained 30 adult males (replicates) for each treatment. 

We dissected all males, extracted their testes, and measured testis volume with the aid of a stereomicroscope (Leica MZ12, Germany) equipped with a digital camera (Olympus SC30, Tokyo, Japan) operated by Olympus CellSens^®^ software (GS-ST-V1.7, Tokyo, Japan). As *E. kuehniella* testes are fused into a spherical organ [19], we calculated its volume using the sphere formula, 4/3πr^3^. We determined the r (radius) using the mean diameter from two measurements across the organ’s central axis divided by two [17,19,64]. After volume measurement, we placed the testes into a drop of Belar saline solution on a cavity slide, tore them apart completely, gently rotated the slide, and counted the number of eupyrene and apyrene sperm under a phase-contrast microscope (Olympus BX51, Tokyo, Japan) according to Liu et al. [19].

### 2.3. Statistical Analysis

Prior to statistical analyses, we fitted data to a general linear model to calculate their residuals and test residual distribution (Shapiro-Wilk test, UNIVARIATE procedure). Data on eupyrene number, apyrene number, and ln(x)-transformed testis size were normally distributed. Because the experimental design was pseudoreplicated, we employed a linear mixed-effects model [65,66] to analyze our data with treatment as a fixed factor and replicate nested into vial (male source) as a random factor [19,66,67,68]. We used a Tukey test for multiple comparisons between treatments. We analyzed the relationship between eupyrenes and apyrenes by a general linear model (GLM procedure) and the slopes of linear lines by an analysis of covariance (ANCOVA) with treatment as the covariate in the model. The numbers of eupyrenes and apyrenes were ln(x)-transformed to achieve normal distribution of data before performing linear regression and ANCOVA. We performed the statistical analyses using SAS 9.4 (SAS, Inc, Cary, NC, USA).

## 3. Results

We demonstrate that males kept in groups (treatments 6M and 1M5F) produced significantly more eupyrene (*F*_2,29_ = 26.31, *p* < 0.0001) and apyrene sperm (*F*_2,29_ = 10.07, *p* = 0.0005) than those maintained singly (treatment 1M) (Figure 2a,b). Sex ratio did not significantly affect production of either eupyrene (*F*_1,29_ = 3.66, *p* = 0.0658) or apyrene (*F*_1,29_ = 3.19, *p* = 0.0847) (Figure 2a,b). Testis size remained similar in all treatments (*F*_2,29_ = 0.01, *p* = 0.9852) (Figure 3). 

Our results show that the ratio of apyrene:eupyrene was about 5:1 with no significant difference between treatments (*F*_2,29_ = 1.24, *p* = 0.3041). The numbers of eupyrenes and apyrenes were significantly positively correlated in all treatments (*F*_1,28_ = 5.31, *p* = 0.0289 for 1M; *F*_1,28_ = 16.65, *p* = 0.0003 for 1M5F; *F*_1,28_ = 11.94, *p* = 0.0018 for 6M) but the slopes of regression lines were not significantly different (*F*_2,84_ = 0.22, *p* = 0.7996) (Figure 4).

## 4. Discussion

We demonstrate for the first time that male pupae of an insect increased sperm production after exposure to conspecific pupal cues regardless of sex (Figure 2). Previous studies report that male insect larvae also can increase their investment in sperm in the presence of non-sex specific larval cues [17,19,20,21]. These findings indicate that juvenile male insects can predict future sperm competition risks from cues of conspecific immature stages and subsequently adjust their sperm production [17,19,69,70,71]. In lepidopteran insects, adults [10] and pupae (current study) adjust production of both fertile eupyrene and infertile apyrene sperm, while larvae only fine-tune production of eupyrene sperm [19] in response to socio-sexual environments. Furthermore, larvae either increase [17,40] or reduce [19] testis size in response to larval cues but pupae (Figure 3) and adults do not change their testis size under different socio-sexual situations. These discoveries suggest that resource allocation to sperm production and testis size differs depending on the life stages exposed to sperm competition environment. 

The above diverse responses to social cues may be attributed to the fact that resource allocation to morphological traits and spermatogenesis takes place in different life stages. Evidence shows that most adult morphological traits are formed during the larval stage [42,43,47,48], allowing the larvae but not pupae and adults to adjust their testis size. Lepidopteran males produce most eupyrene sperm during the larval and pupal stages, most apyrene sperm during the pupal stage [53] and continue to produce both types of sperm during the adult stage [10]. Therefore, male larvae can donate varying portions of testis volumes to spermatogenesis and other functions [41], and trade off testis size and apyrene sperm production to increase eupyrene sperm production in response to increasing sperm competition risk [19]. However, with fully grown testes, adults and pupae have sufficient resources to increase production of both types of sperm in response to sperm competition environment. 

In sperm-heteromorphic insects, the ayprene sperm often overwhelmingly outnumber the eupyrene sperm [12,16,57,72,73]. Previous studies on *E. kuehniella* show that adult males increase the apyrene-eupyrene ratio in response to the presence of rivals [10] but male larvae reduce the ratio after being exposed to larval cues [19]. These may be ascribed to the fact that spermatogenesis of apyrenes and eupyrenes occurs at different stages of insects [53,73] and they have different functions in reproduction [56,57,58,59,60], allowing adults to increase investment in apyrene and larvae to trade-off apyrene for more eupyrene. However, the current study on pupae demonstrates that the apyrene-eupyrene ratio was about 5:1 with no significant difference between treatments. Furthermore, the numbers of eupyrenes and apyrenes were significantly positively correlated in all treatments with no significant difference in the slopes of regression lines (Figure 4). We suggest that in all life stages, males should strive to increase production of eupyrene sperm to ensure advantages in sperm competition (fertilization of more offspring) but also increase production of apyrene when they can (such as at the pupal and adult stages) to ensure successful arrival of eupyrene at the spermatheca. 

Many studies reveal that insect larvae can communicate with each other using non-sex-specific cues [34,35,74,75,76,77,78,79,80,81] and male larvae can adjust their testicular investment in response to these cues [17,19,39,40]. Although female pupae can produce sex pheromones in insects including our study species *E. kuehniella* [35,36,37,38], we have not found any indication that male pupae could respond to this sex specific cue and adjust sperm production accordingly (Figure 2 and Figure 3). Because pupae were in close contact with each other in treatments (2) and (3), physical contact cues could also play a role in pupal response. These findings suggest that testicular investment in *E. kuehniella* juvenile males only responds to the presence of social (including contact), but not sexual cues, during their growth and development. An earlier study demonstrates that *E. kuehniella* adults can remember rival cues and increase sperm allocation for most of their reproductive life after the cues are removed [10]. However, our findings on larval [19] and pupal (current study) responses to social environments result from dissecting adults at emergence. Therefore, we still do not know whether different larval and pupal social exposures influence sperm allocation during their adult lifespan, which warrants further investigation.

## 5. Conclusions

This is the first report on testicular investment in response to the social environment during the pupal stage in an insect. We show that after exposure to pupal cues, male *E. kuehniella* pupae increase production of both eupyrene and apyrene sperm at the same rate but keep testis size unchanged. We suggest that testis size is fixed after pupation because resource allocation to most morphological traits occurs during the larval stage, allowing little room for pupae to adjust testis size. With fully grown testes, pupae can manipulate production of both types of sperm according to the sperm competition risk. Furthermore, sex specific cues such as sex pheromones do not affect sperm production. 

## Figures and Tables

**Figure 1 insects-12-00679-f001:**
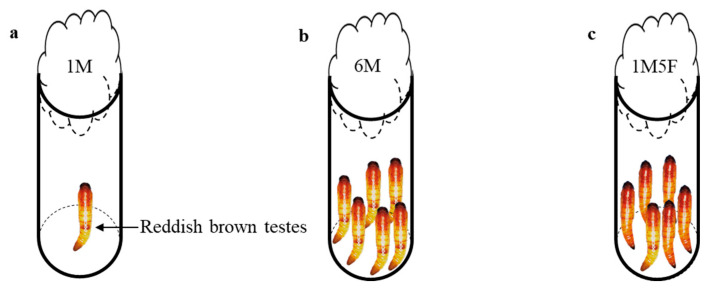
Experimental setup for the entire pupal stage of *E. kuehniella*: (**a**) 1M, one male, (**b**) 6M, six males together, and (**c**) 1M5F, one male and five females together.

**Figure 2 insects-12-00679-f002:**
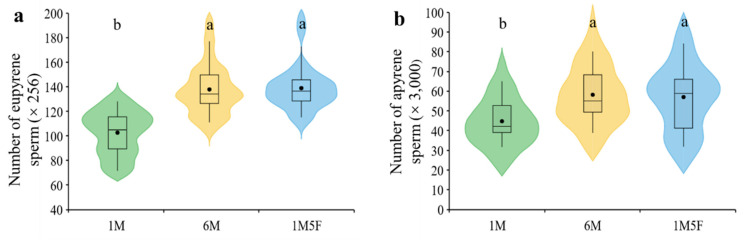
Effect of socio-sexual environment during the pupal stage on the number of eupyrene (**a**) and apyrene (**b**) sperm in testes of *E. kuehniella*. 1M, one male; 6M, six males together; 1M5F, one male and five females together. Each box plot shows the range between the first and third quartiles (black box), mean (black dot) and median scores (black lines); and ‘violin’ shapes show the shape of the distribution. Different letters on the top of the shapes denote significant differences between treatments (*p* < 0.05).

**Figure 3 insects-12-00679-f003:**
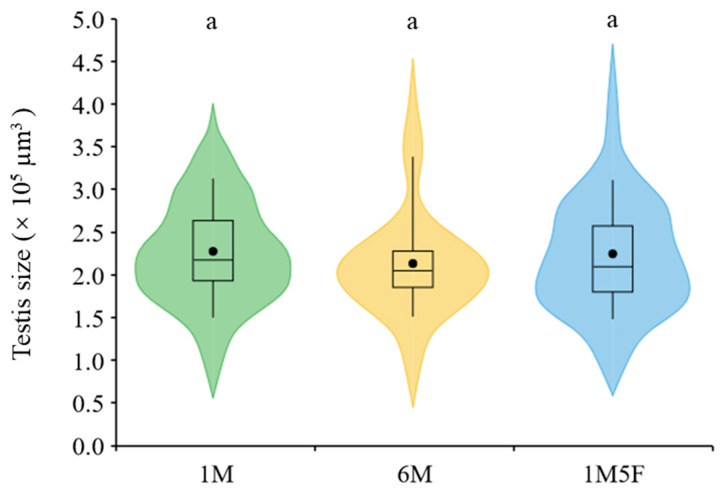
Effect of socio-sexual environment during the pupal stage on testis size of *E. kuehniella*. 1M, one male; 6M, six males together; 1M5F, one male and five females together. Each box plot shows the range between the first and third quartiles (black box), mean (black dot) and median scores (black lines); and ‘violin’ shapes show the shape of the distribution. The same letters on the top of the shapes denote no significant differences between treatments (*p* > 0.05).

**Figure 4 insects-12-00679-f004:**
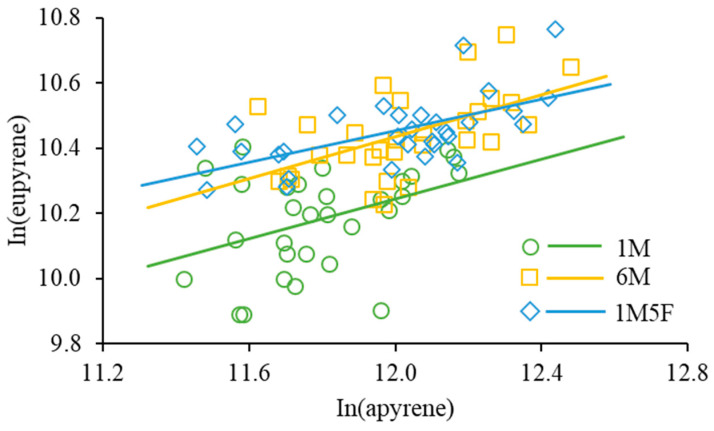
Relationship between the number of eupyrene and apyrene sperm produced. For 1M (one male), ln(eupyrene) = 6.58 + 0.31 × ln(apyrene), R^2^ = 0.1594; for 6M (six males together), ln(eupyrene) = 6.59 + 0.32 × ln(apyrene), R^2^ = 0.2990; and for 1M5F (one male and five females together), ln(eupyrene) = 7.54 + 0.24 × ln(apyrene), R^2^ = 0.3729.

## Data Availability

The datasets are available from the corresponding author upon reasonable request.

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
