# Peer review of "Pupal Cues Increase Sperm Production but Not Testis Size in an Insect"

_insects, 2021, doi:10.3390/insects12080679_

Round 1
Reviewer 1 Report
The manuscript “Pupal cues increase sperm production but not testis size in an insect”, presentes a clear introduction, with updated bibliographic review. This is na analysis to verify if males of Ephestia kuehniella presente testicular investment in reponse to socio-sexula condition, in the pupal stage. The esperimental design is relevant and adequate to the proposed hypothesis. The resultads and discussion are clear and demonstrate the diferente investment of males, in the pupal stage, in which 6 males or 1 male and 5 females were presente in the same environment, in relation to the environment with a single male. In conditions Where there are more individuals regadless of sex, a significantly higher production of eupyrene and apyrene sperm is observed. However, the size of the testicle remains unchanged.
Author Response
We greatly appreciate your positive and constructive comments. We have checked the entire paper again and made minor spelling/grammatic corrections.
Reviewer 2 Report
In their study, the authors found that some stimulation during the pupal stage affects the sperm volume of adult males. There are a number of studies that show that cue in the pupal stage changes traits in the adult stage, as cited in many of the references in this paper. It is difficult to understand the originality of this paper. There is also a problem with the experimental setup, which is a fatal flaw in the discussion. In addition, the cue does not reveal anything at all, and the quality of the research is low. Experiments that reveal at least smell, sound, or contact are common in this type of experiment, and the data is just a count of sperm count, with little content.
Title: Not ‘insect’ but ‘the Mediterranean flour moth Ephestia kuehniella (Zeller)’
L46-47. It is a logical leap to assume that just because female pupae can produce pheromones, male pupae can detect them. Isn't the receiver of the cue an adult as in Duthie et al. (2003)? Pontier & Schweisguth (2015) found communication in the pupa between Drosophila females and males, but what about in this species?
L61. Isn't that what the existing studies show, as in Ref. 19?
L62. Describe the socio-sexual environment in detail.
L98, 114. How did you distinguish the sexes?
L114. Reddish brown testes suddenly appear on the diagram, what is it? Did you sexing with this? Please describe it.
Fig. 2&3. Do you need 'violin' shapes? The variance can be seen to some extent in the box-and-whisker diagram.
L179. The authors did not experiment with different species of pupae, so we do not know if the present results are due to 'same species' pupal cues. It could be explained by density or just mechanical stimulation. You need to experiment with different species, change the density, and redesign some experiment where the density is the same and sex pheromones can reach but mechanical physical stimuli cannot and so on. This data does not sufficiently explain your hypothesis.
L190. In this experiment, the sperm count increased not only in males but also in females in the surrounding area. How can this be explained from the scene of sperm competition? If the males were able to detect the large number of female pupae and increased their sperm count, then this explanation would not apply because it is not sperm competition but a difference in investment strategy.
L203-209. These sentences are `Introduction`.
L209-217. The authors should discuss more about why the ratio of this species remains the same by comparing it with the ecological characteristics of other species you cited.
Reviewer 3 Report
The authors present a straightforward study on the influence of the socio-sexual environment during the pupal phase of one moth species on sperm production. The short manuscript is well written, explains the findings well and contains sound data, which are straightforwardly interpreted.
In general the manuscript merits publication in Insects, however, I have few minor points that should be addressed before publication:
1) keywords: the keyword "sperm production" is already contained in the title and, hence, does not contribute to indexing. It should be substituted by a different one that does so.
2) Material and Methods are well explained, however, there are two sentences, that are ambiguously interpretable and should be slightly specified:
l. 119ff: "We considered all emerged males as replicates, i.e., the male from each 1M vial, the male from each 1M5F vial and all six males from each 6M vial. In total, we obtained 30 adult males (replicates) for each treatment."
Were all six males (6M) considered 1 replicate, or were all emerged males counted as replicates each from the same vial. Please specify. It makes a differences, whether these 30 replicates include 30 individuals (5x6M) or 180 (30 x 6M). If the 6 males from the same vial were considered replicate each, this would be pseudoreplication, as the experimental conditions are the same for each of these 6.
This question also should be tackled in the captions of figure 2 and 3 by adding the number of measurements to the boxes.
l. 126f: "The r (radius) was the mean diameter from two measurements across the organ’s central axis" Was calculated from? please specify, as diameter and radius are two different things.
Furthermore, I would like to suggest to add an image of imagines of the species to the manuscript. But that's optional.
Based on these issues I suggest the acceptance after a minor revision according to the points above. If the measurements on the 6M group include pseudoreplication, this should be solved without it.
Author Response
We greatly appreciate your positive and constructive comments which have improved the paper. We have revised the paper according to these comments. We respond to your specific comments as follows:
- We have changed 'sperm production' into 'spermatogenesis'.
- We considered all males emerged from a vial as replicates. We were aware of pseudoreplication issue and had resolved it in Statistical Analysis in the original submission.
- Radius of the testis: thanks for your comments and correction. We have revised this part accordingly.
- Thanks again for your constructive comments. We have added a colored adult image in Graphic Abstract.
Reviewer 4 Report
This is a manuscript that describes results from an experiment in which pupae of a moth were kept in isolation or with other pupae. The male adults were dissected and size of testes and number of sperm were counted. It was determined that pupae reared in isolation the same size of testes as those reared with other male or female pupae. Male pupae kept with other pupae did increase the number of sperm produced regardless if kept with other male or female pupae. This is an interesting study and I just have a few comments as listed below. Line 55, 56 – change are to were. “sperm were not counted” and “sperm were counted’
Line 103 – unclear why the lid was covered with cloth mesh. Was the lid make of cloth mesh? Unclear what is meant by 2.8 apparatus per mm.
Line 113 – state if in this vial the pupae were in constant contact with each other. Physical contact could be a cue that pupae use to determine density.
Line 145 – In figure 1 it indicates that males have reddish brown testes. Is this how the pupae were sexed? If so indicate in the materials and methods.
Line 223 – another cue that pupae could be using is physical contact.
Author Response
We greatly appreciate your positive and constructive comments which have improved the paper. We respond to your specific comments as follows:
- At line 56 we have changed 'are not counted' into 'were not counted'.
- Excellent comment on line 103. We have revised and clarified this.
- Excellent comment on line 113. We have revised this accordingly. We have also made minor revisions on Discussion to mention contact cues.
- Excellent comment on line 145. We have added the information in M&M.
- Excellent comment on line 223 re contact cues. We have added this information in M&M and Discussion accordingly.